# Vertical Bone Gain after Sinus Lift Procedures with Beta-Tricalcium Phosphate and Simultaneous Implant Placement—A Cross-Sectional Study

**DOI:** 10.3390/medicina56110609

**Published:** 2020-11-13

**Authors:** Juan Manuel Aragoneses Lamas, Margarita Gómez Sánchez, Leví Cuadrado González, Ana Suárez García, Javier Aragoneses Sánchez

**Affiliations:** 1Department of Dental Research, Federico Henriquez y Carvajal University, Santo Domingo 10106, Dominican Republic; jmaragoneses@gmail.com (J.M.A.L.); Javias511@gmail.com (J.A.S.); 2Department of Dentistry, Universidad Europea de Madrid, 28670 Madrid, Spain; 3Department of Implantology and Regeneration, Universidad de Vic-Universidad Central de Cataluña, 08500 Barcelona, Spain; levicuadrado@hotmail.com; 4Department of Pre-clinical Dentistry, Universidad Europea de Madrid, 28670 Madrid, Spain; ana.suarez@universidadeuropea.es

**Keywords:** beta-tricalcium phosphate, vertical bone gain, bone regeneration, maxillary sinus lift, dental implant

## Abstract

*Objectives*: The objective of this cross-sectional study was to evaluate the vertical bone gain achieved after the sinus lift procedure with beta-tricalcium phosphate (β-TCP) used as a bone substitute and simultaneous implant placement. *Methods*: One hundred and twenty-eight sinus lift procedures (utilizing a synthetic ceramic containing 99.9% tricalcium phosphate as a bone substitute) and simultaneous implant placements were performed on 119 patients. The lateral window approach surgical protocol for maxillary sinus lift was performed on the patients. The implants were evaluated using cone-beam computed tomography (CBCT) at 6 months following placement. The vertical bone gain was considered a primary variable, while implant length, diameter, and location were considered secondary variables. *Results*: The ANOVA results showed no statistical difference in vertical bone gain with implant parameters like implant length, width, and position. Pearsons correlation revealed a statistically significant positive correlation with vertical bone gain and implant length and diameter. A further multivariate linear regression analysis was performed and it observed statistically significant associations between the variables in the study after adjusting for confounding factors. *Conclusions*: This study concluded that there was vertical bone gain with the usage of β-TCP in maxillary sinus lift surgical procedure with immediate implant placement and that implant variables like length and diameter had a significant influence on the average bone gain values. The implant position did not have a statistically significant influence but there was considerable variation in the bone gain between first, second premolar, and molar regions.

## 1. Introduction

For more than 30 years, dental implants have proven to be a highly effective and predictable form of treatment modality [1]. However, in the upper jaw, particularly in the posterior area, implants can be a real challenge due to the proximity of the maxillary sinus that can get pneumatized sometimes due to the low bone density in this region [2,3,4,5]. Various treatment options have been proposed to address this problem and they include the use of short implants, placement of angled implants, zygomatic implants, pterygoid implants, as well as maxillary sinus floor elevation [3,4], which is considered as one of the most predictable treatment modality in these situations [6]. A sinus lift can be performed by placing an autogenous, allogeneic, heterologous, or alloplastic graft material [3,5,7]. Initially, autogenous bone grafts were considered to be the “gold standard” material for oral implantology but studies have shown that autografts may not be routinely needed for sinus lift procedures. Additionally, it has various drawbacks like limited availability, increased morbidity of patients, and the unpredictable resorption of the graft that can result in a lack of bone quality and volume [7,8,9,10,11,12].

Kühl et al. [13] stated that the stability of the graft volume appears to be less dependent on the specific graft material (bone substitute) and could be influenced to a greater extent by sinus anatomy and local factors. For these reasons, alternative materials have been sought that provide the same clinical results as autografts, have unlimited availability, avoid the complications of grafting, and are more cost effective. In this context, bioactive ceramics including bioactive glasses, carbonates, sulfates, and calcium phosphates have been touted as possible alternatives for bone regeneration [14]. The present study focused on a calcium phosphate ceramic, Beta-tricalcium phosphate (β-TCP, Ca3(PO4)2), that has been shown to be biocompatible and osteoconductive with a Ca:P ratio similar to that of the mineralized component of bone. It favors bone growth through fixation, proliferation, and differentiation of osteoblasts and mesenchymal cells. [9,15,16,17] Clinically, β-TCP has a rapid rate of degradation compared to other bone substitutes [18,19]. In the existing literature, different studies have observed the combination of β-TCP with autogenous bone [20] or with other biomaterials [21,22,23,24] for improved properties, but the results have been inconclusive [17].

Due to the interesting properties of β-TCP as a unique material to facilitate bone regeneration, several studies have focused on its usage to elevate the floor of the maxillary sinus [25,26]. The purpose of this cross-sectional study was to evaluate the vertical bone gain (radiographic assessment) with the usage of β-TCP graft material in maxillary sinus lift surgery with simultaneous implant placement with a clinical follow-up period of 6 months and also to assess if the implant-related factors have an influence on the achieved bone gain.

## 2. Materials and Methods

Characteristics and settings of the study: This cross-sectional study with a radiographic assessment was conducted at Universidad Federico Henriquez y Carvajal UFHEC (Dominican Republic) from 1 January 2018, to 31 October 2019. The study protocol and findings were reported according to the STROBE guidelines.

Ethics Declaration: This clinical study followed the Declaration of Helsinki on medical protocols and ethics [27]. The protocol was presented to and approved by the Ethics Committee of the Universidad Federico Henriquez y Carvajal UFHEC (Approval number: 10/11/2017) (Approval date: 30 November 2017). The objectives of this study, surgical procedures, materials used, and possible complications of the full protocol were thoroughly explained to the patients, and written informed consent was obtained from all the participants before the start of the study.

Patient Selection: In this study, 119 patients who were indicated for dental implants in the maxillary posterior region were enrolled. The following inclusion criteria were adopted: (i) good overall systemic health; (ii) more than 20 years of age; (iii) need of a prosthetic fixed rehabilitation of unitary elements supported by implants; (iv) presence of edentulous spaces with sufficient keratinized gingiva in the maxilla; (v) insufficient residual bone quantity in the maxillary sinus for implant placement; and (vi) sufficient residual bone quantity in horizontal (more than 4.5 mm from buccal to palatine bone) for the implant diameter. The exclusion criteria included the following: (i) presence of systemic disorders; (ii) sinusitis or other sinus pathologies; (iii) diagnosis of periodontal disease; (iv) bruxism; (v) contraindication for oral surgical procedures; (vi) previous bone augmentation procedures in the maxillary region; (vii) individuals undergoing chemotherapy, radiotherapy or treatment with bisphosphonates; (viii) drug allergies; and (ix) pregnancy.

Surgical Procedure: The pre-treatment bone height measured between the crest of the residual alveolar ridge and the floor of the maxillary sinus was radiologically evaluated using Cone Beam Computed Tomography (CBCT) (Point 3D Combi 500, PoinNix, Ltd., Seul, Korea) (Figure 1).

Among the included patients, 128 maxillary sinus lift surgeries with 260 dental implant placements were performed using the following surgical protocol: The local anesthetic combination of articaine and epinephrine (1:200,000) was administered in the maxillary posterior region. An incision in the mesio-distal axis of the ridge and a vertical discharge incision exceeding the mucogingival line by approximately 5 mm were performed using a no. 11 surgical blade. A full-thickness flap was elevated allowing lateral access to the sinus. The maxillary sinus window was created using a handpiece at 800 rpm and 3 mm tungsten carbide ball milling cutters. Once the design of the window was completed and reflected, the Schneider membrane was lifted using lifting curettes beginning at the lower edge of the antrostomy and ultimately elevating the membrane of the medial and lower walls to allow implant placement. Commercially pure, grade IV titanium dental implants [Frontier^®^ (Global Medical Implants, Lleida, Spain] presenting the same topography were placed at a juxtacrestal level following the guidelines specified by the manufacturer using insertion torque of 25 Ncm. In every intervention, the tooth position, implant diameter, and implant length were documented.

Following implant placement, the sinus cavity was grafted with β-TCP (Iceberg TCP, TCP - GMI Ilerimplant Group) by packing the biomaterial into the floor and the medial wall of the sinus up to the implant apex. The flaps were approximated and sutured using black silk. The post-operative instructions included antibiotics and analgesics for all the patients; 875 mg of amoxicillin and 125 mg of clavulanic acid every 12 h over five days; 600 mg of ibuprofen with arginine every 12 h over three days. The patients were advised to apply 0.2% chlorhexidine topical gel every 8 h for a period of ten days. The sutures were removed after 7 to 10 days if the healing was satisfactory (Figure 2).

In this study, the implants were submerged in the gingival tissue and they were not loaded until after 6 months to facilitate the healing process. At the 6-month follow-up following implant placement and prior to prosthetic loading, a new CBCT was evaluated the residual vertical bone gain measured from the crest of the alveolar ridge to the highest bony point near the apex of the implant and it was subtracted from the pre-treatment bone height. The radiographic examination revealed that the implant apices were completed covered with bone (Figure 3).

Data analysis: The primary outcome parameter of this study was vertical bone gain and analysis was conducted to assess its relationship to implant diameter, length, and tooth position. The secondary outcome parameters included surgical morbidity and implant success rates at 6-month follow-up. The implant success was determined based on criteria proposed by Albrektsson et al. 1986 that included (i) absence of clinical mobility of the implant, (ii) radiographic absence of peri-implant radiolucency, and (iii) absence of infection, pain, neuropathies, or paresthesias [28].

The statistical analysis was performed using Microsoft Excel and IBM SPSS Statistics V.19 software (IBM Corp., Armonk, NY, USA). For descriptive statistical purposes, the common statistical values such as mean, mean confidence interval (99%), standard error, and variance were calculated. The data were summarized in bar charts, boxplot, and tables. The Shapiro-Wilks normality test was performed and the variables followed a parametric distribution. A one-way ANOVA test was performed on the study variables and a DMS post-hoc model was used for multiple comparisons. A significance level of 1% (α = 0.01) was established for the detection of statistically significant differences. Pearson correlation test was performed to assess the correlation between implant variables (length, diameter) with vertical bone gain, and it was followed by multivariate linear regression analysis to assess the strength of each association by adjusting the effect of confounding factors.

## 3. Results

Two hundred and sixty implants with varying diameters (3.75, 4.25, and 4.75 mm) and lengths (10, 11.5, 13, and 15 mm) were placed in the study participants. After six months, in the clinical and radiographic review prior to prosthetic rehabilitation, it was observed that all the surgical sites demonstrated uneventful healing and the implants did not exhibit clinical mechanical looseness, peri-implantitis [29], or fracture during the follow-up period.

The average vertical bone gain obtained was 8.5 ± 0.3 mm per implant following maxillary sinus augmentation with β-TCP. The results from one-way ANOVA showed that there were statistically significant differences between the acquired bone gain for the various implant lengths used in the study (*p*-value < 0.01). Additionally, there were significant differences between the average vertical bone gain and the various implant diameters used in this study (*p*-value < 0.01). While comparing the mean vertical bone gain with implant location, there was no statistically significant difference with a *p*-value of 0.913 (Figure 4). A DMS post hoc analysis for multiple comparisons did not reveal any statistically significant differences between the groups. Though not statistically significant, there was a relative difference in bone gain observed between the locations: (14–15), (14–16), (14–25), and (14–26).

Pearson’s correlation test revealed that there was a statistically significant positive correlation between implant length and average bone gain with a correlation coefficient of 0.51 and a *p*-value of 0.001. There was also a statistically significant positive correlation (correlation coefficient = 0.17) between implant diameter and average bone gain with a *p*-value of 0.004 (Table 1). Multivariate linear regression analysis between the implant variables and mean vertical gain was performed and there was high statistical significance with a *p*-value of 0.001 with both implant diameter and length. For implant length, the regression coefficient value (B) was 0.87 which implied that for every unit increase in implant length, there was a 0.87-fold increase in the average vertical bone gain. A similar finding was observed with implant diameter with B = 1.01 that suggested that for every unit increase in implant diameter, the change in average bone gain was 1.01 times (Table 2).

## 4. Discussion

The presence of reduced bone height in the posterior regions of the maxillary arch can limit the placement of dental implants. The maxillary sinus poses a challenge in the segment of focus and by elevating the floor of the maxillary sinus, with or without graft material as deemed appropriate, an adequate height of bone can be restored for implant placement [30]. It has been observed that the success of a maxillary sinus lift procedure was determined by the amount of new bone formation in the surgical area [25,31,32,33]. A prospective study by Meyer et al., on twenty patients with 33 maxillary sinus lift procedures using β-TCP graft material was conducted over a mean follow-up period of 4.5 years. It was reported that β-TCP induced fewer complications in cases of severe atrophy in the sinus floor and its resorption rate was similar to that of autologous grafts. Additionally, it was observed that the implant success rate was similar to the usage of autografts [34]. In the existing literature, not many studies have investigated the efficacy of β-TCP in maxillary sinus lift procedure and they are usually of smaller sample sizes. The clinical observations from these studies have reported that it is a safe material to be used in this procedure and the obtained results were compared with other types of bone grafts [35,36,37]. Thus, the present study aimed to evaluate the efficacy of β-TCP as a bone substitute in a larger sample set of patients undergoing maxillary sinus lift with immediate implant placement over a follow-up period of 6 months.

The results obtained from the present study observed that β-TCP bone substitute achieved clinical results with substantial vertical bone gain at 6 months in all the study participants where the biomaterial was placed along with immediate implant placement. Ozyuvaci et al., [38] along with other authors [19,39,40], have stated that β-TCP can be reabsorbed and replaced by bone within a short interval of time like six months. This is the primary reason for the utilization of radiographic assessment following 6 months of graft placement in many studies and the same has been followed in the present study [13,20,23,25,31,41,42]. In this study, an average vertical bone gain of 8.5 ± 0.3 mm per implant was observed and it is in concordance with results reported in other studies [43,44]. The bone gain value differed significantly from the data reported by Kiliç et al., [25] where there was a definitive increase in bone gain measurements among a group of subjects treated with β-TCP alone, immediately and six months after the surgical procedure (12.48 ± 2.99 and 11.59 ± 3.02 mm, respectively). Additionally, Meyer et al., observed a mean height gain of 16 mm with this graft material in maxillary sinus lift procedure [34].

In this study, the three-dimensional radiographic assessment using CBCT revealed vertical bone regeneration with β-TCP at 6 months of follow-up and it was dependent on implant variables such as length and diameter. There was a statistically significant positive correlation between implant length, diameter, and average bone gain with β-TCP graft. The confounding factors were adjusted with regression analysis and the strength of the association between the implant variables and average bone gain was statistically significant. There are no studies in the existing literature that has assessed the correlation between implant variables and the outcome parameter such as vertical bone gain with β-TCP graft. A comparable study by Cara-Fuentes et al., performed maxillary sinus lift with filler animal hydroxyapatite (Bio-Oss^®^) and simultaneous implant placement. It reported that there was no correlation between implant length and bone gain at the mesial and distal surfaces of the maxillary sinus [45]. It should be taken into account that during the surgery, β-TCP was placed as much as necessary until all the implants were covered and that could be a plausible reason for longer implants to have obtained greater regeneration when compared to implants of regular length. Although not statistically significant, it should be noted that there was a tendency towards lower bone gain with implants placed in the first premolar region when compared to their counterparts in the second premolar and molar regions. According to Tanaka et al., the possible reason for the lower bone gain in the area of premolars could be because the bone formation begins from the medial and lateral walls [46].

## 5. Conclusions

In the present study, it was observed that vertical bone gain could be achieved with the use of β-TCP in maxillary sinus lift procedure using a lateral window approach along with immediate implant placement over a follow-up period of 6 months. It was also reported that the average bone gain varied between the varying implant lengths and diameter and it had a significant positive correlation with implant variables. The implant position did not have a statistically significant effect on the average bone gain but there was reduced bone gain in the first premolar region when compared to the second premolar and molar regions in the maxillary arch.

## Figures and Tables

**Figure 1 medicina-56-00609-f001:**
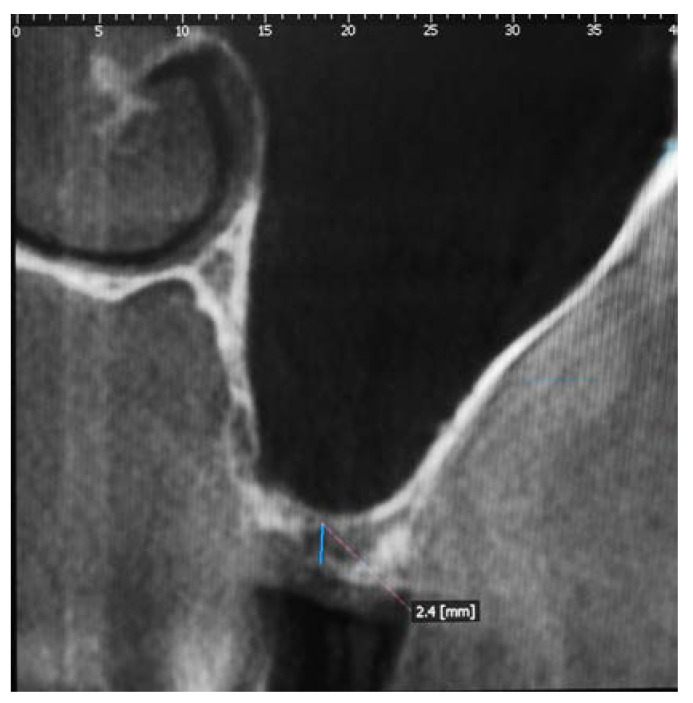
Tomographic section showing the insufficient height of the alveolar ridge.

**Figure 2 medicina-56-00609-f002:**
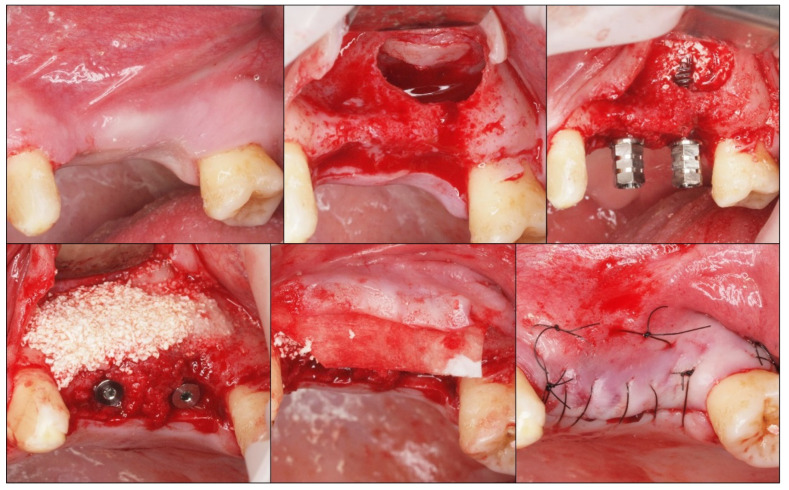
Clinical photographs showing the sequence of sinus lift with the β-TCP graft and the simultaneous placement of the implants.

**Figure 3 medicina-56-00609-f003:**
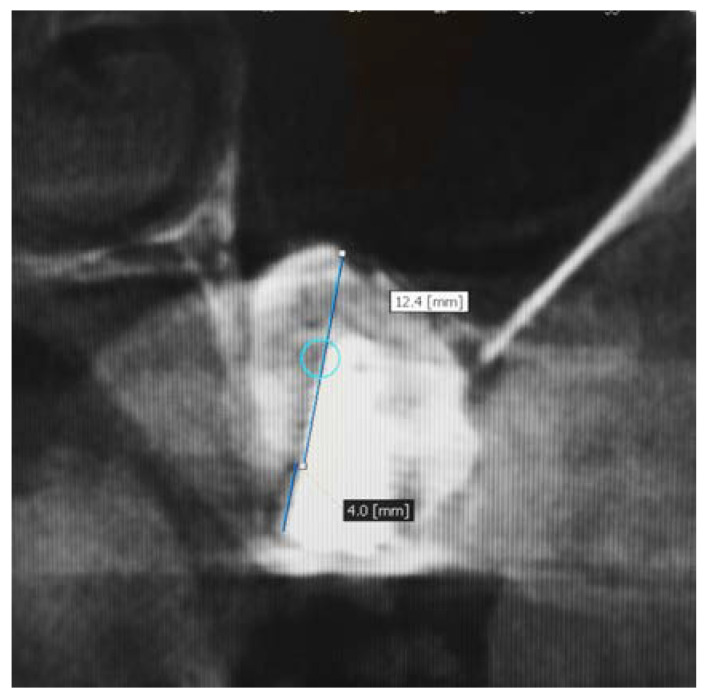
Measurement of bone height at 6 months after insertion of β-TCP and simultaneous placement of implants.

**Figure 4 medicina-56-00609-f004:**
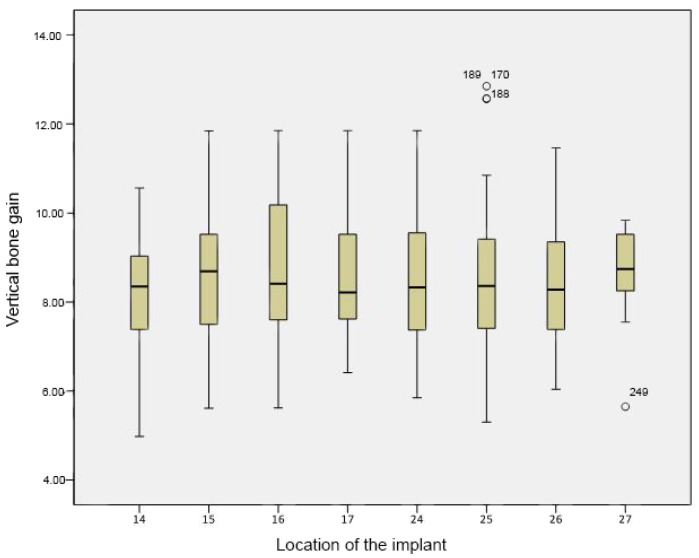
Boxplot of vertical bone gain according to implant location. Vertical bone gain is expressed as a mean, while implant location alludes to the site where the implants were placed, according to the World Dental Federation (FDI) notation system. No statistically significant differences were seen between the average gains (*p*-value < 0.01).

**Table 1 medicina-56-00609-t001:** Pearson Correlation Test between Implant variables and Average Bone Gain.

Implant Variables		Real Bone Gain
Implant Length	Pearson Correlation	0.51
	Sig. (2-tailed)	0.001 **
Implant diameter	Pearson Correlation	0.17
	Sig. (2-tailed)	0.004 **

** *p*-value < 0.001—Highly Statistically significant. Sig.= significance level.

**Table 2 medicina-56-00609-t002:** Multiple Linear Regression Analysis for Implant Variables and Average Bone Gain.

Outcome		Unstandardized Coefficients		Standardized Coefficients	Sig.
		B	Standard Error	Beta	
Implant Variables	Implant Length	0.87	0.08	0.52	0.001 **
	Implant diameter	1.01	0.26	0.20	0.001 **

** *p*-value < 0.001—Highly Statistically significant. Sig.= significance level.

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
