# Peer review of "Vertical Bone Gain after Sinus Lift Procedures with Beta-Tricalcium Phosphate and Simultaneous Implant Placement—A Cross-Sectional Study"

_medicina, 2020, doi:10.3390/medicina56110609_

Round 1

Reviewer 1 Report

I think it would be interesting to explain the prosthetic aspects to understand if that implants have the antagonist or not and if the antagonist was a tooth or prothesis.

Author Response

Dear Sir/Madam,

We sincerely thank the reviewers and the editors of the Medicina for their valuable comments with respect to the manuscript (986496). We have incorporated their inputs and made the necessary amendments in the manuscript. I have attached the response to the reviewers with a detailed description as follows:

REVIEWER 1:

I think it would be interesting to explain the prosthetic aspects to understand if that implants have the antagonist or not and if the antagonist was a tooth or prosthesis.

  1. The authors would like to thank the reviewer for his valuable comment. In this study, the implants were submerged in the gingival tissue following maxillary sinus lift and immediate implant placement. They did not undergo prosthetic loading in order to facilitate the healing process. The antagonistic teeth would also not have an effect on the implants as they were submerged. This point has been clarified in the manuscript in the ‘Materials and Methods’ section Page 4, Para 1, Line 135.

Reviewer 2 Report

In my opinion, it is necessary to detail in words the obtained outcomes in the chapter "Results", especially since they are later presented very briefly and comparatively in the chapter "Discussions".

Author Response

Dear Sir/Madam,

We sincerely thank the reviewers and the editors of the Medicina for their valuable comments with respect to the manuscript (986496). We have incorporated their inputs and made the necessary amendments in the manuscript. I have attached the response to the reviewers with a detailed description as follows:

REVIEWER 2:

In my opinion, it is necessary to detail in words the obtained outcomes in the chapter ‘Results’ especially since they are later presented very briefly and compared in the chapter ‘Discussions’

  1. We heed to the reviewer’s comment and we have given a detailed explanation of the outcome parameters in the results section of the manuscript. Please refer to Page 5, ‘Results’ section, Line 156.

Reviewer 3 Report

The authors describe in their study the augmentation of the maxillary sinus with ß-tricalcium phosphate and simultaneous insertion of implants. The results and conclusions are already known from other studies, but the present study shows data in a very high number of cases, so that the work nevertheless makes a contribution to the scientific literature.

Author Response

Dear Sir/Madam,

We sincerely thank the reviewers and the editors of the Medicina for their valuable comments with respect to the manuscript (986496). We have incorporated their inputs and made the necessary amendments in the manuscript. I have attached the response to the reviewers with a detailed description as follows:

REVIEWER 3:

The authors describe in their study the augmentation of the maxillary sinus with Beta-tricalcium phosphate and simultaneous insertion of implants. The results and conclusions are already known from other studies, but the present study shows data in a very high number of cases, so that the work nevertheless makes a contribution to the scientific literature.

  1. We thank the reviewer for his valuable comment and his recommendation of our study’s findings to the existing literature. This motivates us to perform research on niche topics that can add to the scientific knowledge.

Reviewer 4 Report

Thank you sir for the opportunity of peer-reviewing this manuscript.

Abstract:

In the abstract’s results section the authors are stating that the application of the biomaterial was unrelated to the length of the implant, then in conclusion they are saying that the implant length influenced the vertical bone gain achieved. This is contradictory and it must be rewritten.

Introduction:

Lines 44-48 “ Different studies claim that autogenous bone 44 grafting is considered the "gold standard" procedure” this statement is not true. Autologus bone graft in sinus lifting is not the gold standard anymore for over at least a decade.

Matherial and methods:

Line 77 – please cite the Declaration of Helsinki

Line 111 ´The closing of the mucoperiosteal flap was performed with simple, double-knot stitches with 110 3/0 thread and 3/8 needle, triangular section, half curve, and reverse cut. “ – I don t think this statement is relevant and should be taken out.

Results:

 Statistics should be regression to assess the relationship between the bone height obtained and implant variables . Otherwise, this study does not offer any new material to the existing literature. In this context the authors need to rewrite the methods and the results after multivariate regression statistics is performed.

Discussion:

Line 167 “In fact it has been described that at least a bone height of 8-10 mm is  necessary as a minimum requirement for implant placement” – this statement is not not true. Maybe it was true before sinus lifting procedure invention or short implant placement.

Line 172-178 – is duplicating material presented in the introduction section.

Overall considerations:

English language polishing is mandatory in the whole article, grammar and spelling mistakes can be found in every sentence.

Author Response

Dear Sir/Madam,

We sincerely thank the reviewers and the editors of the Medicina for their valuable comments with respect to the manuscript (986496). We have incorporated their inputs and made the necessary amendments in the manuscript. I have attached the response to the reviewers with a detailed description as follows:

REVIEWER 4:

ABSTRACT:

In the abstract’s results section, the authors are stating that the application of the biomaterial was unrelated to the length of the implant, then in conclusion, they are saying that the implant length influenced the vertical bone gain achieved. This is contraindicatory and it must be rewritten.

  1. We thank the reviewer for his comment and identifying the writing error. We have made the amendments in the abstract section of the edited manuscript. Please refer Page 1, Abstract section, Para 1, Lines 30-40

INTRODUCTION:

Lines 44-48: Different studies claim that autogenous bone 44 grafting is considered the ‘gold standard’ procedure, this statement is not true. Autologous bone graft in sinus lifting is not the gold standard anymore for over at least a decade.

  1. The authors heed to the reviewer’s comment and we have edited the section to highlight the current line of thought that autografts are no longer gold standard graft material for the purpose of maxillary sinus lift procedure. Please find the edit in Page 2, Introduction section, Para 1, Line 53-57.

MATERIAL AND METHODS

Line 77- Please cite the Declaration of Helsinki

  1. The Declaration of Helsinki has been cited in the edited manuscript. Please refer Page 2, Materials and Methods section, Para 5, Line 85.

Line 111- “The closing of the mucoperiosteal flap was performed with simple, double-knot stitches with 110 3/0 thread and 3/8 needle, triangular section, half curve, and reverse cut”- I don’t think this statement is relevant and should be taken out.

  1. We have removed the sentence as per the reviewer’s comment as it was irrelevant.

RESULTS

Statistics should be regression to assess the relationship between the bone height obtained and implant variables. Otherwise, this study does not offer any new material to the existing literature. In this context, the authors need to rewrite the methods and the results after multivariate regression statistics is performed.

  1. The authors have complied to the reviewer’s comments and we have performed Pearson’s correlation test and Multivariate Linear Regression analysis for the implant variables (length and diameter) and average bone gain. The statistical analysis has been added in the data analysis section of Materials and Methods, explained in the Results section, and thoroughly discussed in the Discussion section of the edited manuscript. The results of the analysis have been included in the Conclusion section and it has been highlighted in the relevant sections of the manuscript.

DISCUSSION

Line 167 “In fact it has been described that at least a bone height of 8-10mm is necessary as a minimum requirement for implant placement”- this statement is not true. Maybe it was true before sinus lifting procedure invention or short implant placement.

  1. This line has been removed from the manuscript as per the reviewer’s comment in the edited manuscript.

Line 172-178- is duplicating material in the introduction section.

  1. We thank the reviewer for his/her input and we have removed the duplicating material in the discussion section,

Overall considerations:

English language polishing is mandatory in the whole article, grammar, and spelling mistakes can be found in every sentence.

  1. As per the reviewer’s comment, the manuscript was subjected to English language editing and proof reading professionally. Overall, the manuscript has been amended to incorporate the new statistical results and it has been discussed considering the flow of the manuscript.
